# Multivariate and Geometric Morphometrics Reveal Morphological Variation Among *Sinibotia* Fish

**DOI:** 10.3390/biology14091177

**Published:** 2025-09-02

**Authors:** Yongming Wang, Yong Xie, Yanping Li, Fei Peng, Jinping Li, Wei Jiang, Biwen Xie, Peng Fu, Zuogang Peng

**Affiliations:** 1Key Laboratory of Freshwater Fish Reproduction and Development, Ministry of Education, School of Life Sciences, Southwest University, Chongqing 400715, China; wym8188@126.com; 2Fishes Conservation and Utilization in the Upper Reaches of the Yangtze River Key Laboratory of Sichuan Province, College of Life Science, Neijiang Normal University, Neijiang 641100, China; feipeng2003@126.com (F.P.); ljp040921@126.com (J.L.); xiebw6873@163.com (B.X.); 3Chongqing Fisheries Science Research Institute, Chongqing 400020, China; 13500320866@163.com (Y.X.); 15923096467@163.com (W.J.)

**Keywords:** geometric morphometrics, multivariate morphometrics, phenotypic differences, *Sinibotia* fish, species divergence

## Abstract

*Sinibotia* is a typical representative of the tetraploid lineage of the Botiinae subfamily and is under study to clarify its taxonomic status and phylogenetic relationships. Elucidating phenotypic characteristics is essential for understanding species divergence. In this study, multivariate and geometric morphometric approaches were used to systematically analyze the morphological variations among five *Sinibotia* species. Both methods effectively distinguished between the *Sinibotia* species and yielded highly consistent results. This study emphasizes the effectiveness of the combined use of both methods to obtain comprehensive insights into the morphological differentiation among *Sinibotia* species and their close relationship with ecological adaptation.

## 1. Introduction

Freshwater fish account for approximately 40% of the global fish population and are important components of freshwater ecosystems [1]. Additionally, they are a significant source of protein for humans and are vital to the economy in many countries [2,3]. However, phenomena such as water environment deterioration [4,5], habitat loss [6], overfishing [7,8], biological invasion because of human activities [9,10,11], and global changes [12] pose serious threats to the survival of freshwater fish. The continued increase in the human population threatens the depletion of freshwater fish resources [13]. Consequently, the accurate assessment of resource quantity is the basis for the sustainable utilization of fish resources [14], and the scientific assessment of the stock quantity of various fish depends on the correct identification of species and the effective distinguishing of populations [15,16]. Phenotypic characteristics are important for recognizing organisms and establishing biological classification systems [17]. The response and adaptation of each species to the living environment determine its role and function in the ecosystem [18] and are closely related to the risk of species extinction [19,20]. Therefore, investigating and quantifying the information pertaining to phenotypic characteristics is of great significance for species identification and elucidating the adaptive evolution of freshwater fishes [21,22].

Morphometrics effectively determines the shape and size of organisms, making species identification taxonomically significant [17,23]. Morphometrics rather than molecular markers are widely used in the identification of fish populations because of factors such as low cost, easy operation, and minimal interference from equipment [24,25]. Currently, morphometric studies on species and identification of morphological variation in populations focus primarily on multivariate morphometrics (MM) and geometric morphometrics (GM). MM is a classic method used in fish fauna surveys. It is characterized by strong data traceability and has been widely used to study intra- and interspecific morphological changes in fishery resource management [26,27]. However, MM is based on the linear distance between two points; hence, the spatial pattern of morphological changes is not satisfactorily reflected, which limits its applicability [28]. GM addresses the shortcomings, such as different data sources, non-repeatability, and a lack of separate discussion of size and shape [29,30], and visualizes morphological differences, enables asymmetry assessment, and identifies allometric growth between complex shapes, which are crucial in resolving various issues related to morphological variation [31,32].

Botiidae (Actinopterygii: Cypriniformes) comprises a group of subjectively beautiful small- and medium-sized bottom-dwelling fish that are widely distributed in East, Southeast, and South Asia [33]. These species are of high ornamental and culinary value. The long-term separation between diploid (Leptobotiinae) and tetraploid (Botiinae) lineages of Botiidae presents an excellent model for examining the mechanisms underlying fish polyploidization [34]. The *Sinibotia* genus is a typical representative of the tetraploid lineage of Botiinae and has been subjected to several studies because of its taxonomic status and phylogenetic relationships [35,36,37,38]. Additionally, it is a valuable resource for investigating morphological adaptations and species divergence and gaining insights into patterns of coexistence and interspecific variation owing to the similarity of the *Sinibotia* species with respect to body surface patterns (characterized by broad dark brown bars that extend from one side of the body across the dorsal region to the opposite side and consistently extend below the lateral midline, typically reaching the level of the pelvic fin origin) [38,39] and widespread co-occurrence within the same habitat [38,40].

The recent rapid development of inland fisheries has led to the severe degradation of freshwater fish resources [41,42], and the concurrent commercial spread of these fisheries has led to frequent instances of the appearance of native species in non-native habitats, which poses a challenge for the conservation of *Sinibotia* species [43]. Hence, morphological variations among *Sinibotia* fish species need to be differentiated effectively using comprehensive morphometrics and multivariate statistical analyses. Therefore, the major objectives of this study were: (1) to determine morphometric differentiations using anatomic coordinates and landmark points among five *Sinibotia* species that show similar morphology and close relationships; (2) to provide a geometric topology-based quantitative discriminative system for assessing five *Sinibotia* germplasm resources; and (3) to compare the abilities of MM and GM analyses to capture and distinguish morphological variations in *Sinibotia* species. The overall aim was to provide supplementary information for fishery biology research on *Sinibotia* that could additionally act as foundational data for the assessment and conservation of fishery resources.

## 2. Materials and Methods

### 2.1. Sample Collection

A collection of fish species was performed in August during the years 2017–2023. In total, 150 individuals belonging to five *Sinibotia* species, namely, *S. superciliaris*, *S. reevesae*, *S. robusta*, *S. pulchra*, and *S. zebra,* were collected. Specifically, *S. superciliaris* and *S. reevesae* were obtained from the Tuo River in Zizhong County, *S. zebra* was collected from the Lipu River in Pingle County, and the other two species were collected from the Lijiang River in Pingle County (Figure 1). The specimens were euthanized using MS-222 (Sinopharm Chemical Reagent Co., Ltd, Shanghai, China), weighed, and placed in 95% ethanol (Sinopharm Chemical Reagent Co., Ltd, Shanghai, China) for initial fixation. Subsequently, they were positioned on their left side in a lateral orientation on a Styrofoam plate for photographing using a Canon EOS Kiss X 7 digital camera (Canon, Tokyo, Japan) to obtain a reference scale for standardization during post-processing. During photography, the fish morphology was restored to its natural state and secured using forceps and pins. The camera was mounted on a QH-C082 copystand (Wenzhou Changcheng Movie & TV Equipment Co., Ltd, Wenzhou, China) to maintain the lens parallel to the imaging plane of the specimen. The shooting distance was fixed and maintained at 300 mm to ensure a consistent scale among different samples and reduce the distortion caused by parallax during projection [44]. All procedures were performed by the same researcher to minimize human-induced errors. The specimens were stored in 95% ethanol and deposited at the Fish Conservation and Utilization in the Upper Reaches of the Yangtze River Key Laboratory of Sichuan Province, Neijiang Normal University College of Life Science (Neijiang, China). The samples and morphometric data used in this study are listed in Table 1.

### 2.2. Multivariate Morphometric Analyses

Morphological measurements of 16 traditional morphological traits, such as anal fin length (AFL), body depth (BD), caudal fin length (CFL), caudal peduncle length (CPL), caudal peduncle depth (CPD), dorsal fin length (DFL), eye diameter (ED), head length behind eyes (HBE), head depth (HD), head length (HL), nasal snout distance (nsd), pectoral fin length (PFL), standard length (SL), snout length (SnL), total length (TL), ventral fin length (VFL) were obtained from the photographs of the 150 *Sinibotia* specimens (Figure 2a). Additionally, 10 anatomical coordinates were chosen as truss network traits with the intercoordinate distances designated as D_1–2_ and D_2–3_ (Figure 2b) according to the methodology proposed by Xu et al. [40]. Thus, we obtained 24 comprehensive measurement metrics.

The measurements were captured in pixels using ImageJ (version 1.51j8) software and later transformed into millimeters (accurate to 0.01 mm) by referencing the scale of each image. To minimize measurement errors, two measurements were performed for each sample at different times. All morphological data were standardized by dividing them by the standard length to reduce the influence of length, followed by excluding the standard-length variable [24]. One-way analysis of variance (ANOVA) was used to detect significant differences between all measurements among the *Sinibotia* species, followed by Tukey’s post-hoc test to identify specific differences among these specimens. The values were expressed as mean ± SD. Then, the standardized morphological traits were analyzed using principal component analysis (PCA), discriminant function analysis (DFA), and cluster analysis (CA) using the PAST (version 4.09) software. PCA was performed to decrease data dimensionality and minimize redundancy among specimens. Moreover, the principal components (PCs) were used to build models, and their loadings were extracted for further analysis. DFA was performed to evaluate the accuracy of individual classifications within each population and calculate the success rate of the classification. A stepwise approach was used to identify key features that showed a significant impact. CA was used to analyze multivariate data to group similar populations into common clusters. In fact, a hierarchical cluster analysis was performed based on the Euclidean algorithm (repeated 10,000 times) of morphometric measurements among populations.

### 2.3. Geometric Morphometric Analyses

The external morphological characteristics of the genus *Sinibotia* were used to select 15 homologous landmarks from the lateral view of the body [32], as shown in Figure 3. To precisely delineate the outline of the study subject, 19 semi-landmarks (S1–S19, Figure 3) were generated on the lateral aspect of the body using the sliding technique with the MakeFan (version 8) software from the IMP package. Briefly, three equidistant vertical lines were drawn between each of the remaining pairs of points, with the exception of the segment between points 6 and 7, which was bisected with a single vertical line. The intersection points between the equidistant lines and the lateral profile of the fish acted as semi-landmarks.

The sample photographs were digitally processed using TPS series (tpsUtil version 1.60, tpsDig2 version 2.18, tpsSuper version 2.01) software [32]. The landmarks on the lateral side were arranged in order, starting from the tip of the snout and proceeding clockwise until the intersection point where the end of the occipital bone aligns perpendicularly to the ventral aspect. Subsequently, the landmarks were identified at the most anterior, dorsal, posterior, and ventral points of the orbit and posterior margin of the operculum to ensure consistency in sequence, number, and placement of markers on each image. All procedures were performed by the same researcher to minimize potential errors due to human variability.

After digitization, the data were imported into the MorphoJ (version 1.07) software for Generalized Procrustes Analysis (GPA) [45] to eliminate the effects of non-morphological variations caused by changes in position, orientation, and scale during the selection of landmark points. This is to ensure optimal results for subsequent statistical analysis. PCA was used to assess the geometric variation across samples. Canonical variate analysis (CVA) was used to assess morphological variations across *Sinibotia* populations, whereas DFA was used to compare morphological differences within pairwise populations. Quantitative analysis of morphological differences between pairs of populations was performed using Mahalanobis distance, followed by 10,000 repetitions of *p*-value testing. The Mahalanobis distance quantifies the covariance distance in the data, which aids in assessing the dissimilarity of an individual from other individuals in the sample and indicates the extent of overlap in inter-individual morphological variation. Additionally, the TPS files with the landmark information for each group were processed using tpsSuper (version 2.01) to derive the average shapes of each group, which were combined using tpsUtil (version 1.60) and imported into the PAST (version 4.09) software for cluster analysis based on the Euclidean algorithm (repeated 10,000 times) to elucidate the variations among all PCs.

## 3. Results

### 3.1. Morphological Variation Based on Multivariate Morphological Metrics

After transformation, the correlation coefficient between the standard length and 39 size-independent morphometric characteristics notably decreased. Initial data indicated correlation coefficients ranging from 0.336–0.804; the most transformed values showed correlations of <0.6 after adjustment, with the exception of D_2–9_. Among the specimens, *S. reevesae* was the longest, followed by *S. superciliaris*, whereas *S. zebra* was the shortest. Subsequent multivariate analysis showed that the ratio of specimen number to morphometric variables (N:P) was 3.85 [46].

One-way ANOVA showed significant variations in all 39 morphometric variables in all five *Sinibotia* species (*p* < 0.05, Appendix A). As indicated in Table 2, with the exception of head length behind eyes (HBE), dorsal fin length (DFL), anal fin length (AFL), D_2–8_, D_2–9_, D_4–5_, D_4–6_, D_4–10_, and D_5–6_, the remaining 30 morphological indices exhibited two extremes in both the *S. robusta* and *S. zebra* specimens. Specifically, *S. robusta* showed the highest values for eye diameter (ED), head depth (HD), ventral fin length (VFL), caudal peduncle length (CFL), total length (TL), D_1–2_, D_2–10_, D_3–4_, D_3–7_, D_3–8_, D_4–9_, D_7–8_, and D_8–9_ (*p* < 0.05), whereas *S. zebra* showed the highest values for caudal peduncle length (CPL), D_2–3_, D_3–10_, D_5–7_, D_6–7_, and D_9–10_ (*p* < 0.05). *S. pulchra* showed the largest HBE and D_4–5_ (*p* < 0.05), whereas *S. superciliaris* showed similarities to *S. reevesae*. Significant differences (*p* < 0.05) were noted in various morphological parameters, such as snout length (SnL), nasal snout distance (NSD), HD, body depth (BD), AFL, CFL, DFL, caudal peduncle depth (CPD), TL, D_2–3_, D_2–8_, D_2–9_, D_3–4_, D_3–8_, D_3–9_, D_3–10_, D_4–5_, D_4–8_, D_4–9_, D_4–10_, D_5–6_, D_5–8_, and D_9–10_.

PCA identified six components with eigenvalues >1 from the morphometric measurements, which accounted for 82.10% of the variations among the five *Sinibotia* fish species (Appendix A). Specifically, the first and second components contributed 45.22% and 13.84% of the variation, respectively. The PCA scatter plot (Figure 4a) showed that *S. zebra* exhibited significant morphological differences from those of the other four species and could be completely separated using the anatomical landmarks chosen in this study. In contrast, *S. superciliaris* shares morphological similarities with *S. reevesae*, *S. pulchra*, and *S. robusta* and cannot be entirely distinguished through PCA analysis. Furthermore, DFA based on MM analysis showed effective differentiation between *S. zebra*, *S. pulchra*, and *S. robusta*, whereas *S. superciliaris* and *S. reevesae* were difficult to distinguish and showed morphological overlaps (Figure 4b).

The three PCs with the highest contribution rates were selected for the factor analysis (Table 3). PC1 showed strong correlations with most morphological variables, with the exception of HBE, NSD, D_1–2_, D_2–8_, D_2–9_, D_3–10_, D_4–6_, D_4–10_, D_5–8_, and D_9–10_. The most robust correlations were observed with indices related to body depth (BD, D_3–7_, D_3–8_, D_3–9_, and D_4–9_). PC2 exhibited a negative correlation with D_1–2_ and a positive correlation with all other variables, with strong correlations observed with HL, CFL, TL, D_1–2_, D_2–3_, D_2–8_, D_2–9_, D_3–10_, and D_9–10_. PC3 primarily correlated with DFL, AFL, and D_4–10_.

Further analysis showed that BD, CFL, DFL, HBE, HD, PFL, SnL, D_3–4_, D_4–5_, D_4–9_, D_4–10_, D_5–6_, and D_5–7_ exhibited the highest loadings among the 39 morphological traits and significantly influenced the discriminant analysis. The classification function coefficients for each population based on these key variables are listed in Table 4. Furthermore, stepwise discriminant analysis showed 100% comprehensive discriminant rates and cross-validation rates for each group (Appendix A), which indicates a precise discrimination of the five *Sinibotia* species.

Euclidean distances were calculated for the hierarchical cluster analysis using the average value of each morphometric measurement for all five species. As shown in Figure 5, categorization of the five *Sinibotia* species showed two primary clades: one comprising *S. pulchra* and *S. zebra*, and the other formed by the initial clustering of *S. superciliaris* with *S. reevesae*, followed by their subsequent grouping with *S. robusta*.

### 3.2. Morphological Variation Based on Geometric Morphometrics

The *Sinibotia* species dataset, which comprised 34 landmarks in lateral view, showed that PC1 and PC2 explained 56.96% and 14.40% of the variance, respectively (Figure 6a). The primary factors that influenced trunk height variation were landmarks 3 and 9, along with the semi-landmarks S6 and S14, which were associated with PC1. In contrast, landmarks 2 and 5, which were linked to PC2, primarily explained the variability in head height. The PCA of 34 landmark points in the lateral view showed a partial morphological overlap between *S. pulchra* and *S. zebra*; however, they could be distinguished from *S. superciliaris*, *S. reevesae*, and *S. robusta*. *S. superciliaris* and *S. reevesae* showed significant morphological overlap with each other, whereas *S. robusta* exhibited minor overlap with *S. superciliaris* and *S. reevesae*.

CVA showed that CV1 and CV2 represented 64.00% and 26.79% of the total variance, respectively (Figure 6b). The scatter plot graph with CV1 and CV2 as the X and Y axes showed a distinct separation among *S. robusta* populations with 95% confidence. In contrast, *S. pulchra* and *S. zebra* populations were in close proximity, and some overlap was noted between the *S. superciliaris* and *S. reevesae* populations. Analysis of the CV1 axis showed that *S. pulchra* and *S. zebra* were situated on the negative side of the axis, whereas *S. robusta* predominantly occupied the positive side. *S. superciliaris* and *S. reevesae* were positioned at the center of the axis. The morphological variations that showed positive values on the CV1 axis included shortened head, enlarged eyes, increased trunk height, and shortened yet heightened tail (Figure 6b). Analysis of the CV2 axis showed that *S. robusta*, *S. pulchra*, and *S. zebra* were distributed along the negative axis, whereas *S. superciliaris* and *S. reevesae* were positioned along the positive axis (Figure 6b). Morphological alterations that showed positive values on the CV2 axis were shortened and reduced head, decreased eye size, elongated trunk, and shortened tail (Figure 6b).

The DFA based on 34 landmarks of the *Sinibotia* species corroborated the findings of PCA and CVA, which highlight the distinctiveness of *S. robusta*. The DFA showed that *S. robusta* proximity was between *S. pulchra* and *S. zebra*, and it overlapped between *S. superciliaris* and *S. reevesae* (Appendix A). DFA was performed on pairwise combinations of the five *Sinibotia* species to determine the phenotypic differentiation trends between the populations. According to the Mahalanobis distance results (Figure 7), the morphological differences were highly significant among the *Sinibotia* populations (*p* < 0.001), and all groups achieved a 100% discriminant rate. Among them, *S. robusta* showed the largest Mahalanobis distance compared with those of the other populations, particularly *S. zebra* (21.6623). In contrast, *S. pulchra* exhibited a relatively small Mahalanobis distance from *S. zebra*, whereas *S. superciliaris* and *S. reevesae* showed the shortest distance between them (6.7297). These results emphasize the significant morphological variation between *S. robusta* and *S. zebra*, whereas *S. superciliaris* and *S. reevesae* showed the least variability.

The thin-plate spline and warped outline drawings included in Figure 7 show notable morphological disparities in the head, trunk, and tail regions of the five *Sinibotia* species. Observation of head morphology showed that *S. zebra* exhibited the shortest head length, and *S. robusta* and *S. pulchra* showed the longest. *S. superciliaris* and *S. reevesae* showed intermediate lengths. Notably, *S. robusta* exhibited the tallest head, followed by *S. reevesae*, whereas *S. superciliaris* and *S. pulchra* showed similar head heights, and *S. zebra* showed the shortest head height. Additionally, *S. robusta* exhibited the largest eyes, whereas *S. zebra* showed the smallest eyes. The snouts showed similar positioning in *S. superciliaris* and *S. reevesae*, although they were notably elevated compared to those of the other *Sinibotia* species. Study of trunk morphology showed that trunk height varied in the following order: *S. robusta* > *S. reevesae* > *S. superciliaris* > *S. pulchra* > *S. zebra*, which was the smallest in stature. Analysis of the tail region showed that *S. pulchra* and *S. zebra* showed the longest and shortest caudal peduncles, respectively. The other three *Sinibotia* species exhibited similar caudal peduncle lengths, with *S. reevesae* and *S. robusta* having comparable and relatively tall caudal peduncle lengths.

Based on the CA, the five *Sinibotia* species were generally categorized into two groups: *S. pulchra* and *S. zebra* were grouped together, whereas *S. superciliaris* was first placed with *S. reevesae*, and both were placed with *S. robusta* (Appendix A). This result was consistent with the results of the MM analysis of the 39 morphometric variations.

## 4. Discussion

### 4.1. Benefits of Combining Multivariate and Geometric Morphometrics in Species Identification

The identification of species and their stocks within natural populations is essential for estimating global fish diversity and is crucial for open-water fishery management and the sustainable use of fish species for human welfare [47,48,49]. Although genetic tools are highly effective and widely used in the conservation and management of fishery resources [50], an excessive genetic focus may detract from the goals of taxonomy, systematics, and population characterization. This highlights the need for a holistic approach [51]. MM and GM are techniques that are commonly used for species identification—particularly for fish classification [52], individual ontogeny [53], population subdivision [54], and physiological ecology [55]. These distinct morphological measurement approaches often emphasize the different dimensions of morphological variation among individuals or taxonomic groups; thus, they capture diverse aspects of morphology, which provide varying interpretations of morphological differences in fish [56]. Hence, the integration of diverse morphometric measurement techniques is essential to achieve a comprehensive analysis of the morphological variations in fish species or populations. Studies similar to the current one have been conducted on the morphological identification of the *Chimarrichthys* fish complex [57], coral reef fish [56], and sardines [58].

This present study integrated MM and GM to hierarchically investigate the morphological differences among five *Sinibotia* species. Both methods yielded results that indicated that the morphological differences among the *Sinibotia* species phenotypes were primarily concentrated in head length and depth, eye size, trunk depth, tail length, and height. Additionally, multivariate statistical analyses consistently indicated the relative independence of *S. robusta*, whereas *S. pulchra* and *S. zebra* were grouped together, owing to their similar morphologies. *S. superciliaris* and *S. reevesae* exhibited minimal morphological distinctions and certain overlap. These results are consistent with those of previous studies on the phylogeny of botiid loaches using mitochondrial and nuclear genes [59], which validates the efficacy of both approaches in differentiating *Sinibotia* species. Furthermore, this study has highlighted two distinct strategies that have led to slightly varying interpretations of the morphological differences in *Sinibotia* species in fin and body morphology. GM effectively captured shape information from homologous landmarks by visually interpreting the morphological and size differences among the five *Sinibotia* species; however, MM methods provided a more accurate quantification of the differences in fish body shape. Hence, morphological analysis using GM methods needs to be considered in tandem with MM analysis, and the essential contribution of MM should not be dismissed when using GM for morphological analysis.

### 4.2. Morphological Variation Among Sinibotia Species

The morphological structure of an organism is the result of the prolonged interplay of genetic information and environmental factors [60], and it not only acts as a primary basis for species identification and classification [61], but also plays a crucial role in the study of biological evolution [62], adaptive mechanisms [63], and functional ecology [64]. The *Sinibotia* genus was first established by Fang in 1936 [65], and it includes multiple species that have been described and documented to date [39]. The predominant distribution of *Sinibotia* in biodiversity hotspots in East Asia necessitates its accurate quantification and thorough understanding of its morphological variations, not only to form the basis of taxonomic and systematic studies, but also to represent a critical pathway for identifying ecological adaptations, evolutionary processes, and functional trade-offs.

Morphological studies on *Sinibotia* fish have been limited to basic descriptions of certain species. Bohlen et al. [38] conducted a comparative study of 33 morphological characteristics and reported that *S. zebra* showed a morphology that was intermediate to those of *Leptobotia guilinensis* and *S. pulchra*. They noted a resemblance between the head stripes of *S. zebra* and *S. pulchra*, whereas the body marking pattern closely resembled that of *L. guilinensis*. Xu et al. [40] compared 10 traditional morphological parameters and 20 truss network features of *S. superciliaris* and *S. reevesae* and found that PCA could not differentiate between the two species. Wu et al. [39] have indicated that snout length, eye diameter, body depth, suborbital spine shape, and body coloration are important criteria for classifying and identifying species in the *Sinibotia* genus. Therefore, by combining the MM and GM methods, the present study has identified more diverse morphological differences than those reported by previous studies. Based on the overall findings, the five *Sinibotia* species exhibited significant differences in head length (SnL and HBE), HD, ED, HD, CPL, CPD, and fin morphology (CFL, DFL, and PFL). The most pronounced differences in morphology were observed between *S. robusta* and *S. zebra*, which showed substantial variations in multiple morphological indicators. In contrast, the morphological differences between *S. superciliaris* and *S. reevesae* were the least significant. The morphological variations among the five *Sinibotia* species were as follows: *S. robusta* exhibited the highest HD, ED, CFL, D_3–4_, and D_4–9_; *S. pulchra* showed the highest HBE and D_4–5_; and *S. zebra* showed the highest CPL and D_5–7_. Notable differences between *S. superciliaris* and *S. reevesae* included variation in SnL, BD, CFL, DFL, CPD, D_3–4_, D_4–9_, D_4–10_, and D_5–6_.

### 4.3. Morphological Variation and Ecological Adaptation Among Sinibotia Species

Fish form the largest group of vertebrate species; hence, the diversity in fish morphology is the result of long-term adaptation to complex and ever-changing aquatic environments. Incidentally, this highlights the significance of natural selection in shaping biological phenotypes [66]. The common ancestor of the *Sinibotia* genus likely began to diverge in the Early Miocene (21.1 Mya) and spread from the mainland of Southeast Asia to East Asia [67]. The geological movements in East Asia since the Miocene period (such as the uplift of the Qinghai–Tibet Plateau) and changes in water systems have enabled the species to develop distribution patterns that are suitable for habitation in the Pearl River (*S. robusta*, *S. pulchra*, and *S. zebra*) and Yangtze River systems (*S. superciliaris* and *S. reevesae*) [59]. Investigating the morphological differentiation and speciation of *Sinibotia* in allopatric or sympatric regions could provide insights into the mechanisms that maintain species diversity and provide additional theoretical guidance for the sustainable utilization of fishery resources and species conservation.

*Sinibotia* fish in the Pearl River system showed three morphs: *S. pulchra* showed similarity to *S. zebra*, and *S. robusta* was distinct. In contrast, the *Sinibotia* fish in the Yangtze River system showed two morphs that fell between the three morphs found in the Pearl River system. Both ancient connections and significant genetic differentiation are known to exist among the fish populations in the Yangtze and Pearl River systems, which aligns with the speculated ancient hydrological evolution history [68]. This leads to the speculation that the ancestors of the *Sinibotia* genus were the first to enter the Pearl River system but later became isolated because of the evolution of the water system (such as river capture, channel changes, and ancient water system fragmentation) and glacial–interglacial cycles, leading to gradual differentiation into different species under different selection pressures and genetic drift [69,70,71].

The significant variation in morphology among the three *Sinibotia* species found in the Pearl River system could be attributed to the distinct geographic, topographic, and climatic conditions of the basin, which offer a diverse range of ecological niches for fish within the basin [72]; furthermore, this could have influenced the adaptive evolution of their morphology. *S. robusta* exhibits characteristics that show adaptation to slow-flowing or still water [73]. The larger eyes are conducive to the quick spotting of prey and predators [74], and its robust body and long pectoral, pelvic, and caudal fins ensure high maneuverability, which is essential for flexibility in swimming in complex environments, maintaining stability, and rapid escape [75,76]. *S. pulchra* and *S. zebra* possess small eyes, slender bodies, narrow caudal peduncles, and short fins, which help reduce resistance during swimming and are suitable for fast and sustained swimming or living in swift water currents [77]. Furthermore, despite their notable morphological similarities, *S. superciliaris* and *S. reevesae* did display variations that may be attributed to habitat utilization. For example, the body shape of *S. superciliaris* is slender, whereas that of *S. reevesae* is robust. This indicates a strong affinity for interspecific hybridization between *S. superciliaris* and *S. reevesae* [78]. However, recent overfishing [79], increased water pollution [80], and hydraulic engineering [81] have significantly increased the competition for limited resources and potential hybridization risk between *S. superciliaris* and *S. reevesae*, which exist under the same habitat conditions, posing challenges for the conservation of the species. Thus, further study of the morphological variance among closely related species of the *Sinibotia* genus, with a specific focus on the interaction of multiple ecological factors, phenotypic plasticity, and mechanisms underlying morphological differentiation and species formation, is crucial for enhancing our understanding of the ecological adaptability and morphological differentiation patterns of *Sinibotia* species. Furthermore, this study provides essential theoretical evidence for the conservation and sustainable utilization of aquatic biodiversity.

## 5. Conclusions

This study has successfully identified significant morphological differences among five *Sinibotia* species through the integrated application of multivariate and geometric morphometrics. These differences were primarily observed in head length, head depth, body depth, caudal peduncle length, and caudal peduncle depth. These findings not only provide important morphological evidence for species identification and phylogenetic relationships within the *Sinibotia* genus, but also offer a new perspective for understanding its underlying adaptive evolutionary mechanisms. Furthermore, to address the high conservation pressures for *Sinibotia* species, future studies should explore a broader geographical spectrum of morphological variations by incorporating osteology, comparative genomics, and epigenetics to comprehensively elucidate the molecular mechanisms underlying speciation and adaptive evolution of the *Sinibotia* species.

## Figures and Tables

**Figure 1 biology-14-01177-f001:**
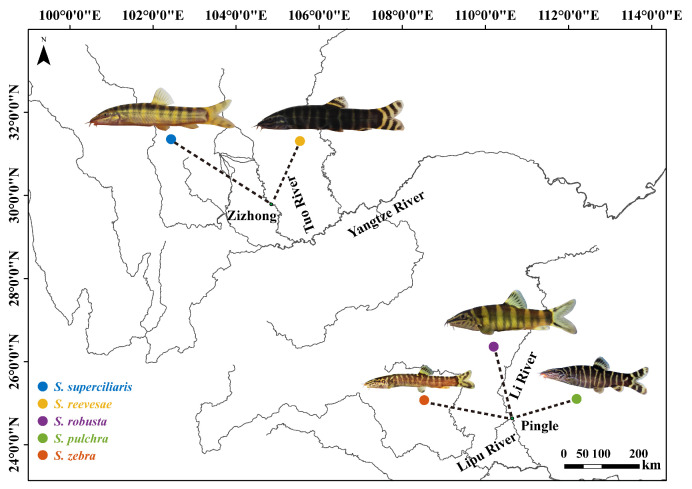
Map of sampling sites of *Sinibotia* fishes. *S. superciliaris* and *S. reevesae* were collected from the Tuo River in Zizhong County in August 2017. The other three species were collected from the Li and Lipu Rivers in Pingle County in August 2023.

**Figure 2 biology-14-01177-f002:**
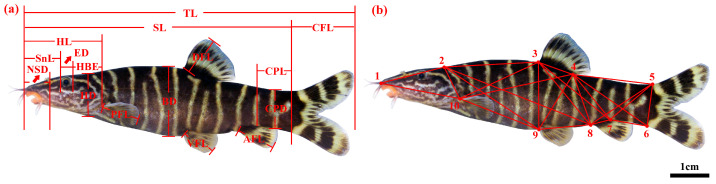
Multivariate morphometric metrics measured. (**a**). Initially, 16 traditional morphological characteristics were measured: AFL, anal fin length; BD, body depth; CFL, caudal fin length; CPL, caudal peduncle length; CPD, caudal peduncle depth; DFL, dorsal fin length; ED, eye diameter; HBE, head length behind eyes; HD, head depth; HL, head length; NSD, nasal snout distance; PFL, pectoral fin length; SL, standard length; SnL, snout length; TL, total length; VFL, ventral fin length; (**b**). Additionally, 24 variables were extracted from 10 anatomic coordinates: 1. tip of snout; 2. end of occipital bone; 3. origin of dorsal fin base; 4. end of dorsal fin base; 5. dorsal origin of caudal fin base; 6. ventral origin of caudal fin base; 7. end of anal fin base; 8. origin of anal fin base; 9. origin of pelvic fin base; and 10. origin of pectoral fin base; D_1–2_ denotes the distance between anatomic coordinates 1 and 2.

**Figure 3 biology-14-01177-f003:**
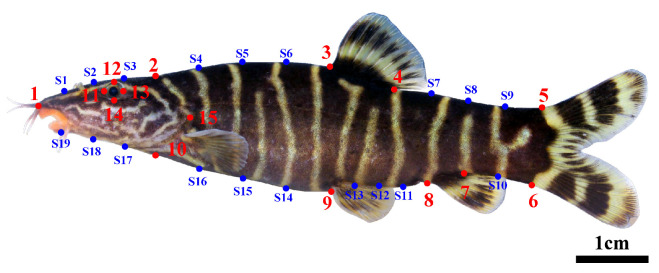
Landmarks and semi-landmarks used for geometric morphometric analysis. 1. tip of snout; 2. end of occipital bone; 3. origin of dorsal fin base; 4. end of dorsal fin base; 5. dorsal origin of caudal fin base; 6. ventral origin of caudal fin base; 7. end of anal fin base; 8. origin of anal fin base; 9. origin of pelvic fin base; 10. point of intersection of the end of the occipital bone perpendicular to the ventral aspect; 11. anterior-most point of orbit; 12. dorsal-most point of orbit; 13. posterior-most point of orbit; 14. ventral-most point of orbit; 15. posterior-most margin of operculum. S1–19 indicate the intersection points between the equidistant lines and the lateral profile of the fish. The red dots indicate homologous landmarks, and blue dots indicate semi-landmarks.

**Figure 4 biology-14-01177-f004:**
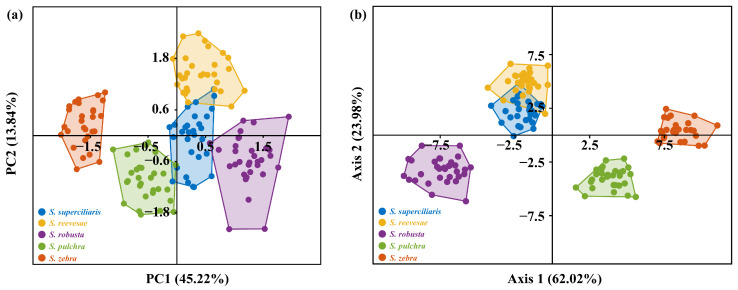
Scatter plot for multivariate morphological characters of five *Sinibotia* species. (**a**) Result of principal component analysis from PC1 and PC2, and (**b**) discriminant function analysis plot with 39 morphometric variations.

**Figure 5 biology-14-01177-f005:**
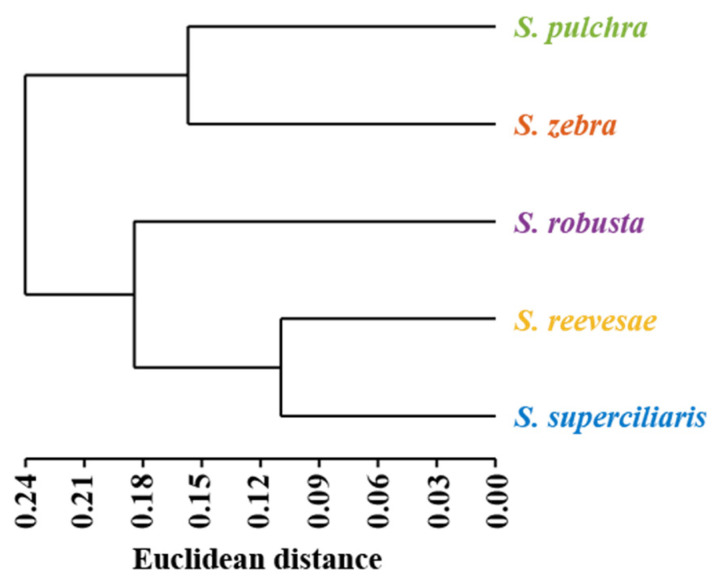
Dendrogram based on multivariate morphometric measurements.

**Figure 6 biology-14-01177-f006:**
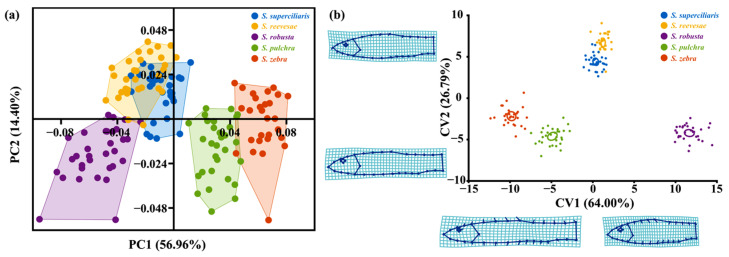
Scatter plot of geometric morphometric analysis of five *Sinibotia* species. (**a**) Principal component analysis of PC1 and PC2, and (**b**) canonical variate analysis of the five *Sinibotia* species based on 34 landmarks on the lateral view. Circles in CVA indicate 95% confidence.

**Figure 7 biology-14-01177-f007:**
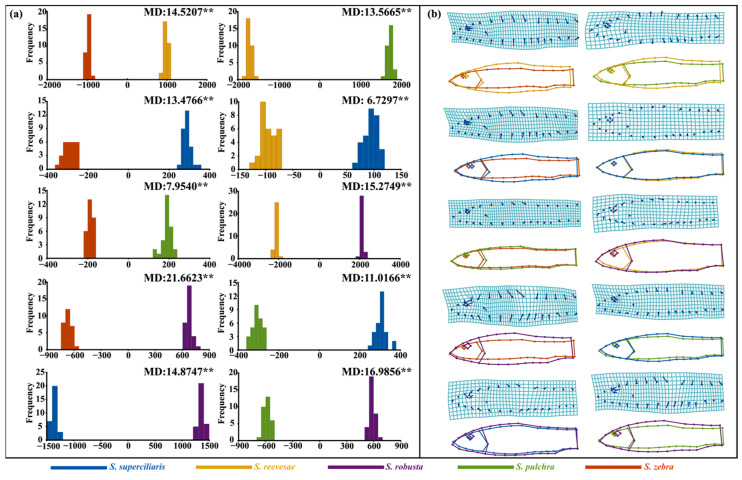
Relative distortion analysis of the *Sinibotia* species based on 34 landmarks on the lateral view. Discriminant scores (**a**) and warped outline drawings of the morphological structure (**b**) between populations. Discriminant function analyses results are indicated with Mahalanobis distances (MD) on the upper right of the discriminant score plots. Significance levels: ** *p* < 0.001.

**Table 1 biology-14-01177-t001:** Sample and morphometric information of five *Sinibotia* fishes used in this study.

Species	*n*	Sampling Location	Standard Length (mm)		Total Weight (g)	
			Range	Mean ± SD	Range	Mean ± SD
*S. superciliaris*	30	Zizhong, Tuo River	76.95–81.33	79.14 ± 5.87	6.87–7.96	7.41 ± 1.46
*S. reevesae*	30	Zizhong, Tuo River	89.38–95.39	92.38 ± 8.05	12.64–15.29	13.96 ± 3.54
*S. robusta*	32	Pingle, Li River	69.06–74.49	71.77 ± 7.54	7.57–9.22	8.40 ± 2.29
*S. pulchra*	30	Pingle, Li River	73.04–77.28	75.16 ± 5.69	4.78–5.51	5.14 ± 0.98
*S. zebra*	28	Pingle, Lipu River	66.34–70.57	68.46 ± 5.45	2.93–3.45	3.19 ± 0.67

**Table 2 biology-14-01177-t002:** Description statistics for morphological characters of five species in *Sinibotia* (Mean ± SD).

Variable	*S. superciliaris*	*S. reevesae*	*S. robusta*	*S. pulchra*	*S. zebra*
HL	0.269 ± 0.011 ^b^	0.263 ± 0.014 ^b^	0.281 ± 0.012 ^a^	0.278 ± 0.013 ^a^	0.245 ± 0.010 ^c^
ED	0.031 ± 0.006 ^bc^	0.036 ± 0.009 ^b^	0.053 ± 0.011 ^a^	0.029 ± 0.008 ^cd^	0.025 ± 0.009 ^d^
SnL	0.117 ± 0.010 ^a^	0.108 ± 0.013 ^b^	0.114 ± 0.009 ^ab^	0.111 ± 0.009 ^ab^	0.096 ± 0.007 ^c^
HBE	0.119 ± 0.009 ^b^	0.119 ± 0.009 ^bc^	0.113 ± 0.012 ^c^	0.137 ± 0.009 ^a^	0.124 ± 0.006 ^b^
HD	0.158 ± 0.009 ^c^	0.166 ± 0.008 ^b^	0.194 ± 0.012 ^a^	0.153 ± 0.012 ^c^	0.131 ± 0.012 ^d^
NSD	0.070 ± 0.008 ^a^	0.061 ± 0.009 ^b^	0.070 ± 0.009 ^a^	0.070 ± 0.009 ^a^	0.059 ± 0.007 ^b^
BD	0.215 ± 0.015 ^b^	0.237 ± 0.015 ^a^	0.241 ± 0.019 ^a^	0.166 ± 0.013 ^c^	0.140 ± 0.013 ^d^
DFL	0.164 ± 0.023 ^a^	0.147 ± 0.025 ^b^	0.139 ± 0.013 ^bc^	0.126 ± 0.013 ^c^	0.101 ± 0.010 ^d^
PFL	0.142 ± 0.024 ^ab^	0.135 ± 0.022 ^b^	0.152 ± 0.020 ^a^	0.103 ± 0.013 ^c^	0.066 ± 0.012 ^d^
VFL	0.121 ± 0.017 ^b^	0.119 ± 0.016 ^b^	0.134 ± 0.016 ^a^	0.104 ± 0.012 ^c^	0.077 ± 0.012 ^d^
AFL	0.146 ± 0.019 ^a^	0.132 ± 0.021 ^b^	0.133 ± 0.014 ^b^	0.111 ± 0.010 ^c^	0.088 ± 0.012 ^d^
CPL	0.152 ± 0.006 ^bc^	0.159 ± 0.010 ^b^	0.149 ± 0.013 ^c^	0.160 ± 0.013 ^b^	0.175 ± 0.019 ^a^
CPH	0.144 ± 0.007 ^b^	0.164 ± 0.010 ^a^	0.158 ± 0.011 ^a^	0.130 ± 0.009 ^c^	0.126 ± 0.012 ^c^
CFL	0.222 ± 0.026 ^b^	0.199 ± 0.032 ^c^	0.305 ± 0.027 ^a^	0.234 ± 0.012 ^b^	0.178 ± 0.014 ^d^
TL	1.221 ± 0.024 ^b^	1.197 ± 0.031 ^c^	1.289 ± 0.021 ^a^	1.230 ± 0.016 ^b^	1.172 ± 0.012 ^d^
D_1–2_	0.222 ± 0.014 ^c^	0.218 ± 0.016 ^c^	0.254 ± 0.016 ^a^	0.239 ± 0.016 ^b^	0.219 ± 0.020 ^c^
D_1–10_	0.272 ± 0.012 ^a^	0.262 ± 0.017 ^a^	0.272 ± 0.016 ^a^	0.272 ± 0.014 ^a^	0.243 ± 0.016 ^b^
D_2–3_	0.336 ± 0.017 ^b^	0.361 ± 0.016 ^a^	0.303 ± 0.023 ^c^	0.330 ± 0.022 ^b^	0.368 ± 0.021 ^a^
D_2–8_	0.598 ± 0.015 ^bc^	0.620 ± 0.023 ^a^	0.601 ± 0.021 ^b^	0.585 ± 0.019 ^c^	0.583 ± 0.023 ^c^
D_2–9_	0.392 ± 0.017 ^b^	0.419 ± 0.017 ^a^	0.378 ± 0.016 ^c^	0.369 ± 0.014 ^c^	0.377 ± 0.017 ^c^
D_2–10_	0.147 ± 0.011 ^b^	0.152 ± 0.013 ^b^	0.170 ± 0.017 ^a^	0.132 ± 0.012 ^c^	0.113 ± 0.011 ^d^
D_3–4_	0.126 ± 0.009 ^c^	0.134 ± 0.011 ^b^	0.177 ± 0.011 ^a^	0.122 ± 0.009 ^c^	0.103 ± 0.011 ^d^
D_3–7_	0.368 ± 0.015 ^b^	0.375 ± 0.016 ^b^	0.414 ± 0.015 ^a^	0.348 ± 0.012 ^c^	0.300 ± 0.016 ^d^
D_3–8_	0.313 ± 0.023 ^c^	0.325 ± 0.013 ^b^	0.358 ± 0.015 ^a^	0.295 ± 0.014 ^d^	0.251 ± 0.014 ^e^
D_3–9_	0.212 ± 0.016 ^b^	0.237 ± 0.016 ^a^	0.237 ± 0.018 ^a^	0.163 ± 0.016 ^c^	0.140 ± 0.013 ^d^
D_3–10_	0.321 ± 0.016 ^b^	0.360 ± 0.02 ^a^	0.327 ± 0.014 ^b^	0.327 ± 0.016 ^b^	0.354 ± 0.012 ^a^
D_4–5_	0.306 ± 0.014 ^b^	0.292 ± 0.012 ^c^	0.285 ± 0.018 ^c^	0.324 ± 0.013 ^a^	0.310 ± 0.015 ^b^
D_4–6_	0.347 ± 0.014 ^ab^	0.349 ± 0.013 ^ab^	0.348 ± 0.013 ^ab^	0.356 ± 0.013 ^a^	0.343 ± 0.017 ^b^
D_4–7_	0.257 ± 0.011 ^a^	0.257 ± 0.012 ^a^	0.261 ± 0.013 ^a^	0.231 ± 0.009 ^b^	0.204 ± 0.013 ^c^
D_4–8_	0.217 ± 0.011 ^b^	0.227 ± 0.011 ^a^	0.232 ± 0.014 ^a^	0.190 ± 0.012 ^c^	0.169 ± 0.013 ^d^
D_4–9_	0.218 ± 0.010 ^c^	0.244 ± 0.018 ^b^	0.256 ± 0.017 ^a^	0.179 ± 0.015 ^d^	0.169 ± 0.014 ^d^
D_4–10_	0.423 ± 0.012 ^c^	0.466 ± 0.023 ^a^	0.467 ± 0.017 ^a^	0.433 ± 0.015 ^bc^	0.441 ± 0.011 ^b^
D_5–6_	0.147 ± 0.007 ^c^	0.170 ± 0.011 ^a^	0.155 ± 0.010 ^b^	0.125 ± 0.011 ^d^	0.121 ± 0.009 ^d^
D_5–7_	0.111 ± 0.013 ^c^	0.117 ± 0.011 ^c^	0.117 ± 0.014 ^c^	0.142 ± 0.013 ^b^	0.153 ± 0.013 ^a^
D_5–8_	0.250 ± 0.012 ^b^	0.271 ± 0.011 ^a^	0.271 ± 0.015 ^a^	0.257 ± 0.016 ^b^	0.251 ± 0.012 ^b^
D_6–7_	0.111 ± 0.013 ^c^	0.117 ± 0.011 ^c^	0.117 ± 0.014 ^c^	0.142 ± 0.013 ^b^	0.153 ± 0.008 ^a^
D_7–8_	0.081 ± 0.008 ^b^	0.081 ± 0.009 ^b^	0.091 ± 0.008 ^a^	0.078 ± 0.010 ^b^	0.065 ± 0.008 ^c^
D_8–9_	0.223 ± 0.017 ^b^	0.220 ± 0.018 ^bc^	0.244 ± 0.010 ^a^	0.225 ± 0.013 ^b^	0.212 ± 0.012 ^c^
D_9–10_	0.292 ± 0.014 ^b^	0.322 ± 0.021 ^a^	0.285 ± 0.016 ^b^	0.297 ± 0.018 ^b^	0.316 ± 0.013 ^a^

Note: AFL, anal fin length; BD, body depth; CFL, caudal fin length; CPL, caudal peduncle length; CPD, caudal peduncle depth; D_i–j_ denotes the distance between anatomic coordinates i and j according to Figure 2b; DFL, dorsal fin length; ED, eye diameter; HBE, head length behind eyes; HD, head depth; HL, head length; NSD, nasal snout distance; PFL, pectoral fin length; SnL, snout length; TL, total length; VFL, ventral fin length; different letters in superscript indicate significant differences (*p* < 0.05).

**Table 3 biology-14-01177-t003:** Loading of the first three components in PCA based on multivariate morphological characters.

Variable	Principle Component			Variable	Principle Component		
	PC1	PC2	PC3		PC1	PC2	PC3
HL	0.620	−0.508	−0.050	D_2–10_	0.858	−0.007	−0.029
ED	0.675	−0.079	0.281	D_3–4_	0.844	−0.175	0.328
SnL	0.550	−0.323	−0.087	D_3–7_	0.954	−0.051	0.092
HBE	−0.404	−0.304	−0.256	D_3–8_	0.930	0.010	0.083
HD	0.876	−0.141	0.134	D_3–9_	0.909	0.315	0.002
NSD	0.317	−0.437	−0.055	D_3–10_	−0.217	0.709	0.390
BD	0.914	0.293	−0.028	D_4–5_	−0.553	−0.261	−0.187
DFL	0.560	0.131	−0.666	D_4–6_	0.062	−0.199	0.125
PFL	0.818	0.043	−0.378	D_4–7_	0.858	0.116	−0.172
VFL	0.781	−0.011	−0.332	D_4–8_	0.898	0.203	−0.063
AFL	0.665	0.093	−0.573	D_4–9_	0.900	0.270	0.179
CPL	−0.585	0.064	0.178	D_4–10_	0.441	0.423	0.695
CPD	0.773	0.381	0.067	D_5–6_	0.759	0.414	−0.019
CFL	0.699	−0.502	0.342	D_5–7_	−0.726	−0.203	0.371
TL	0.713	−0.504	0.314	D_5–8_	0.425	0.171	0.343
D_1–2_	0.404	−0.637	0.367	D_6–7_	−0.726	−0.203	0.371
D_1–10_	0.538	−0.370	−0.131	D_7–8_	0.746	−0.158	0.088
D_2–3_	−0.512	0.704	−0.008	D_8–9_	0.521	−0.207	0.393
D_2–8_	0.410	0.630	0.164	D_9–10_	−0.309	0.662	0.228
D_2–9_	0.283	0.828	−0.095				

Note: AFL, anal fin length; BD, body depth; CFL, caudal fin length; CPL, caudal peduncle length; CPD, caudal peduncle depth; D_i–j_ denotes the distance between anatomic coordinates i and j according to Figure 2b; DFL, dorsal fin length; ED, eye diameter; HBE, head length behind eyes; HD, head depth; HL, head length; NSD, nasal snout distance; PFL, pectoral fin length; SnL, snout length; TL, total length; VFL, ventral fin length.

**Table 4 biology-14-01177-t004:** Classification function coefficients of the *Sinibotia* species based on multivariate morphological characters.

Variable	*S. superciliaris*	*S. reevesae*	*S. robusta*	*S. pulchra*	*S. zebra*
BD	−649.906	−652.521	−593.096	−842.953	−992.365
CFL	855.276	740.269	1032.533	891.593	681.982
DFL	1936.75	1732.945	1686.195	1905.229	1783.063
HBE	1564.229	1575.658	1506.593	1960.871	1880.49
HD	1235.458	1267.257	1753.985	1485.928	1320.554
PFL	91.896	135.38	313.923	−35.366	−154.035
SnL	3564.3	3229.94	3105.582	3576.356	3316.018
D_3–4_	625.357	564.064	1202.921	710.819	411.203
D_4–5_	3524.526	3330.814	3443.542	3590.367	3329.115
D_4–9_	−962.795	−882.986	−1146.886	−1194.772	−973.426
D_4–10_	3206.205	3305.072	3233.779	3432.547	3425.339
D_5–6_	−412.321	−76.278	−458.97	−701.59	−532.633
D_5–7_	−342.912	−273.204	−233.37	−70.455	68.72
Constent	−1693.693	−1669.88	−1817.476	−1813.591	−1622.784

Note: BD, body depth; CFL, caudal fin length; D_i–j_ denotes the distance between anatomic coordinates i and j according to Figure 2b; DFL, dorsal fin length; HBE, head length behind eyes; HD, head depth; PFL, pectoral fin length; SnL, snout length.

## Data Availability

The data presented in this study are available upon request from the corresponding author.

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
