# Peer review of "Multivariate and Geometric Morphometrics Reveal Morphological Variation Among *Sinibotia* Fish"

_biology, 2025, doi:10.3390/biology14091177_

Round 1

Reviewer 1 Report

Comments and Suggestions for Authors

Overall, the manuscript entitled, “Multivariate and geometric morphometrics revealed morphological variation among Sinibotia fish” is a commendable piece of research with important implications for comprehensive insights into the morphological differentiation among Sinibotia species and their close relationship with ecological adaptation. However, some sections could benefit from additional clarity and more concise language to enhance readability. The structure of the article is logical, with a clear progression from introduction to features and examples. The tone is generally informative, which suits the purpose of the article. Hence, the manuscript can be accepted after major revisions as follows:

- A graphical abstract will make the manuscript more interesting to the reader.

- Please consider more recent publications (2023-2025) in the manuscript.

- Arrange the keywords in an alphabetic order.

- Please add the ethical certificate number of the study.

- Line 53: add references.

-Line 143: Mention the 16 traditional morphological traits selected for the morphological measurements.

- Line 166: Mention how the values were expressed, wherever mean ± SD or mean ± SE.

- Under table 1: add a caption about how the values were expressed.

- In fig2: you must add a ruler under the fish from the beginning of the body showing fish total length.

- Under table 2: write the full names of all abbreviations mentioned.

- A clear explanation is required in Material and methods section about how to perform a Dendrogram based on multivariate morphometric measurements.

Reviewer 2 Report

Comments and Suggestions for Authors

Dear Authors, 

In this research, multivariate and geometric morphometric features were used to reveal morphological variation among Sinibotia fish. The results are clearly and concisely explained using appropriate methodology. They were evaluated and supported by the literature. Minor corrections are presented in the attached file. The material and methods section could be shortened, particularly by removing commonly known information. This paper is acceptable for publication with minor revisions. All corrections are marked in the PDF file.

Reviewer 3 Report

Comments and Suggestions for Authors

The manuscript titled “Multivariate and geometric morphometrics reveal morphological variation among Sinibotia fish” examines the morphological differences among five species from China rivers. Main questions are whether these species have different (or similar) morphology. Additionally, in the manuscript there are two different morphological methods (multivariate morphometric analyses) and geometric morphometric analyses in order to examine whether these methods produce the same results. Using these methods the manuscript seeks to establish a macroscopically discriminative method for their distinction. The topic is quite relevant to the concept of the Journal, whereas it scratches the surface of a modern issue, about the growing demand on food supply due to human overpopulation. Moreover, it examines the species' biodiversity that are not well described or examined. The manuscript is well structured, well written and gives all the information, even for a reader that is not well informed about Sinobiota species. It answers the aims that were established in the introduction, throughout the manuscript from material and methods, up to the discussion. The methods are quietly explained with all the information needed. Both methods are examined thoroughly and all the results are presented and discussed sufficiently. Finally, in the end of the discussion section, the manuscript gives additional information about future studies, giving the impression that this is the first step of a study. I believe that this manuscript should be published in its present form with a single revision as mentioned below:

(L180) “Three major landmark points”: Since there are 15 landmarks used for the GM analysis, the above expression confuses the readers. Therefore, this part should be explained furtherly, otherwise, the whole sentence should be rephrased.

Author Response

Comments 1: (L180) “Three major landmark points”: Since there are 15 landmarks used for the GM analysis, the above expression confuses the readers. Therefore, this part should be explained furtherly, otherwise, the whole sentence should be rephrased.

Response 1: Thank you for your comment and valuable opinion. To avoid ambiguity, we have modified this section as “The external morphological characteristics of the genus Sinibotia were used to select 15 homologous landmarks from the lateral view of the body [32] , as shown in Figure 3. ” Meanwhile, the modified parts in the manuscript have been marked in red. Please see more details in the lines 186-187.

Round 2

Reviewer 1 Report

Comments and Suggestions for Authors

I recommend the publication of the article after checking the major revisions submitted by authors. The CORRECTED copy of the manuscript has been
sufficiently improved to warrant publication in Biology.